# Evaluation of a Cardiovascular Systems Model for Design and Analysis of Hemodynamic Safety Studies

**DOI:** 10.3390/pharmaceutics15041175

**Published:** 2023-04-07

**Authors:** Yu Fu, Nelleke Snelder, Tingjie Guo, Piet H. van der Graaf, Johan. G. C. van Hasselt

**Affiliations:** 1Leiden Academic Centre for Drug Research, Leiden University, Einsteinweg 55, 2333 CC Leiden, The Netherlands; 2LAP&P Consultants BV, Archimedesweg 31, 2333 CM Leiden, The Netherlands; 3Certara QSP, Canterbury CT2 7FG, UK

**Keywords:** hemodynamics, systems pharmacology, cardiovascular, pharmacodynamics, modeling

## Abstract

Early prediction, quantification and translation of cardiovascular hemodynamic drug effects is essential in pre-clinical drug development. In this study, a novel hemodynamic cardiovascular systems (CVS) model was developed to support these goals. The model consisted of distinct system- and drug-specific parameter, and uses data for heart rate (HR), cardiac output (CO), and mean atrial pressure (MAP) to infer drug mode-of-action (MoA). To support further application of this model in drug development, we conducted a systematic analysis of the estimation performance of the CVS model to infer drug- and system-specific parameters. Specifically, we focused on the impact on model estimation performance when considering differences in available readouts and the impact of study design choices. To this end, a practical identifiability analysis was performed, evaluating model estimation performance for different combinations of hemodynamic endpoints, drug effect sizes, and study design characteristics. The practical identifiability analysis showed that MoA of drug effect could be identified for different drug effect magnitudes and both system- and drug-specific parameters can be estimated precisely with minimal bias. Study designs which exclude measurement of CO or use a reduced measurement duration still allow the identification and quantification of MoA with acceptable performance. In conclusion, the CVS model can be used to support the design and inference of MoA in pre-clinical CVS experiments, with a future potential for applying the uniquely identifiable systems parameters to support inter-species scaling.

## 1. Introduction

Cardiovascular toxicity remains one of the most significant causes of market withdrawal [1,2,3]. In this context, pre-clinical and clinical cardiovascular safety studies and associated quantitative analyses have primarily focused on early identification of QT prolongation effects, associated with an increased risk of torsade de pointes. However, drugs may also be associated with other non-QT cardiovascular hemodynamic drug effects; such effects are now increasingly being considered in cardiovascular safety studies [4]. To assess these hemodynamic drug effects, studies in animal models, such as dog and rat, are commonly conducted while measuring readouts, such as heart rate (HR), mean arterial pressure (MAP), and cardiac output (CO) or cardiac contractility. One key purpose of such pre-clinical studies is identifying the specifical hemodynamic endpoint on which the drug is acting, i.e., the mode-of-action (MoA). Due to the complex regulatory system which drives the cardiovascular hemodynamic response and the subsequent relationship between observed changes in hemodynamic readouts, identifying the MoA in pre-clinical cardiovascular studies is not simple [5]. In addition, data from such pre-clinical studies should ideally be used to predict hemodynamic drug effects in other species, including humans; however, it is hampered by potential inter-species differences.

Quantitative mathematical models may have potential to support the optimization of experimental designs of pre-clinical and clinical hemodynamic studies. In past studies, a quantitative hemodynamic cardiovascular systems model (CVS) was developed using experimental hemodynamic study data for multiple drugs in rats, including measurements of HR, CO, and MAP [6,7]. This CVS model characterizes short-term drug effects on the inter-relationship between five hemodynamic variables, i.e., HR, stroke volume (SV), total peripheral resistance (TPR), CO, and MAP. The model allows incorporation of drug effects associated with these variables to evaluate and quantify the likely drug mode-of-action (MoA). The use of distinct drug- and system-specific parameters in the model may also enable inter-species scaling of drug effects towards other species, such as dogs [8] and humans [9]. As such, the CVS model has potential value to support classification of the drug MoA and the quantification of its effect size, supporting pre-clinical study design optimization; it could also produce translational cross-species predictions, such as supporting dose selection in first-time-in-human studies. The CVS model was developed based on experimental data for a selected set of compounds with specific characteristics. Broader application of the CVS model is currently limited by a lack of general quantitative characterization of the model’s performance in terms of correct identification and unbiased estimation of hemodynamic drug effects; any broad application must take into account study design considerations, such as the effect of differences in the type of available hemodynamic readouts, sample size, drug doses, and the magnitude of the effect size.

In this study, we describe a systematic analysis of the estimation performance of the CVS model to infer drug- and system-specific parameters, evaluating its utility to support the design and analysis of pre-clinical cardiovascular studies. Specifically, we aimed to evaluate the following elements for the CVS model: (1) the practical identifiability of drug- and system-specific parameters; and (2) the estimation performance in terms of correct identification and parameter estimation precision in relation to drug effects, available hemodynamic readouts, and experimental study design characteristics.

## 2. Methods

### 2.1. Model Structure

The hemodynamic systems model used in this analysis was developed by Snelder et al. [7]; it will hereafter be referred to as the CVS model. In summary, the CVS model includes the inter-relationship of five hemodynamic variables, including HR, SV, CO, TPR, and MAP (Figure 1). CO is defined as the product of HR and SV, while MAP is the product of CO and TPR. The drug effect is described with an indirect response model reflecting the impact of drug on the turnover of primary hemodynamic biomarkers HR, SV, or TPR, while CO, MAP, and HR data were used as readouts for model estimation. The previously described handling effect was excluded from this analysis since its magnitude is much smaller than the magnitude of circadian rhythm or drug effect. The differential equations of the model are as follows:$$\frac{dHR}{dt}=K_{in\_HR}\cdot (1+{CR}_{HR})\cdot \left(1-FB*MAP\right)\cdot (1+EFF)-K_{out\_HR}\cdot HR$$
$$\frac{d{SV}^{*}}{dt}=K_{in\_SV}\cdot \left(1-FB*MAP\right)\cdot (1+EFF)-K_{out\_SV}\cdot {SV}^{*}$$
$$\frac{dTPR}{dt}=K_{in\_TPR}\cdot (1+{CR}_{TPR})\cdot \left(1-FB*MAP\right)\cdot (1+EFF)-K_{out\_TPR}\cdot TPR$$
$$SV={SV}^{*}\times [1-HR_{SV}\times LN\left(\frac{HR}{BSL_{HR}}\right)]$$
CO=HR·SV
MAP=CO·TPR
$$CR_{HR}=amp_{HR}\cdot cos(2\pi \cdot \frac{t+hor_{HR}}{24})$$
$$CR_{TPR}=amp_{TPR}\cdot cos(2\pi \cdot \frac{t+hor_{TPR}}{24})$$
$$EFF=\frac{E_{max}\cdot C}{{EC}_{50}+C}$$

The model equations consist of system-specific parameters (i.e., feedback-, turnover-, and circadian rhythm-related parameters) and drug-specific parameters (i.e., EC_50_ and E_max_). The FB constant represents the negative feedback of MAP on the turnover of HR, SV, and TPR. The parameters K_in_HR_, K_in_SV_, and K_in_TPR_ are the zero-order production constants for HR, SV and TPR, while K_out_HR_, K_out_SV_, and K_out_TPR_ are the first-order dissipation constant for HR, SV and TPR, respectively. BSL is the baseline value of each hemodynamic variable. Circadian rhythms (CR) for HR and TPR were modelled with a cosine function, as previously described. For the CR equations, amp and hor represent amplitude and horizontal displacement, respectively. HR_SV_ is a constant to describe the magnitude of direct effect of HR and SV. Regarding pharmacokinetics (PK), we assumed a hypothetical drug with the PK described according to a one-compartment model. The elimination rate constant (K) of the hypothetical drug was set as 0.173 h^−1^ corresponding to a realistic half-life of 4 h. A typical E_max_ model was assumed to link the drug exposure to the drug effect (EFF) on either HR, SV, or TPR. C represents the drug concentration at the central compartment. Drug-specific parameters EC_50_ and E_max_ represent the concentration at which half of the maximum and maximum effects are achieved, respectively. Inter-individual variability was included on baseline parameters (BSL) for HR, CO and MAP. The full model code of the model is included in Appendix A.

### 2.2. Simulation Scenarios

All simulation scenarios, including study design characteristics and model parameters used, are summarized in detail in Appendix A; the scenarios included one scenario for the quantification of system- and drug-specific parameters and three scenarios for identification of the MoA with different drug effects, different observation duration, or different numbers of animals. We briefly simulated dosing regimens for animals receiving either (i) three ascending doses: 0.1 mg/kg, 1 mg/kg, and 10 mg/kg *i.v.* bolus on day 1, 2, and 3; or (ii) five ascending doses: 0.1 mg/kg, 0.3 mg/kg, 1 mg/kg, 3 mg/kg, and 10 mg/kg *i.v.* bolus on days 1 to 5. These dosing regimens follow commonly applied study designs. We adopted a dense sampling design for the simulations, taking hourly measurements after drug administration of HR, CO, and MAP as, in these studies, rich sampling using telemetry is common practice. For all the scenarios focusing on identification of the MoA, drug-specific parameters (E_max_ and EC_50_), interindividual variability (IIV), and residual errors of each type of measurement were estimated; the system-specific parameters were fixed to the parameter values from the published CVS model [7].

### 2.3. Practical Identifiability Analysis

#### 2.3.1. Stochastic Stimulation and Re-Estimation

The practical identifiability analyses of the CVS model for identification of drug- and systems parameters, and the evaluation of drug effect, available readouts, and study designs, were all performed using stochastic simulations and re-estimation (SSE) analyses. SSEs were performed using Perl-speaks-NONMEM toolkit (PsN version 4.8.1, Uppsala University, Uppsala, Sweden) in conjunction with NONMEM (version 7.4.3, Icon Development Solutions, Ellicott, MD, USA). The SSE is a commonly used tool for model comparison and hypothesis testing. In SSE analyses, a number of simulated datasets were generated using the input model (original model). The simulated datasets were then used to fit the original model and a set of alternative models. Ultimately, bias and precision of parameter estimates, as well as statistical significance of tested models, can be computed. A classical first-order conditional estimation with interaction (FOCE-I) method was used for model estimation.

#### 2.3.2. Identifiability of Drug- and System-Specific Parameters

We evaluated whether the CVS model could be used to identify drug- and system-specific parameters based on common hemodynamic study designs. For this assessment, we implemented CVS model structures that included drug effect parameters on one of the three primary hemodynamic variables (HR, TPR, and SV) and a model which did not include a drug effect on hemodynamics as the alternative model. We first simulated a number of datasets using the models with drug effects on either HR, SV, or TPR, respectively (original models). We then re-fitted the simulated datasets using the original (i.e., true) models, as well as alternative models with either drug effect on the other hemodynamic variables (i.e., incorrect models) or no drug effect included at all. For each scenario, we performed 100 simulations and used them to fit both original models and alternative models (re-estimations). For each re-estimated model, we evaluated the precision of parameter estimates using the relative prediction error (RPE), calculated as follows:$$RPE\left(\%\right)=\frac{P_{est}-P_{sim}}{P_{sim}}\times 100\%$$
with P*_est_* being the re-estimated parameter value and P*_sim_* the original, simulated value.

To assess the ability of each model to identify the correct drug effect, we calculated the decrease in objective function value (ΔOFV) between the base and the alternative models for each simulation scenario (Figure 2). We classified a significant drug effect as a decrease in OFV > 5.99 (*p* < 0.05, degree of freedom = 2 in chi square test), compared to the model with no drug effect.

#### 2.3.3. Drug Effect Magnitudes

We performed SSEs with either an inhibitory or stimulatory drug effect on HR, SV, or TPR to confirm if the MoA could be identified correctly for different drug effect magnitudes and directionalities. The E_max_ in the original models was set to 1 for inhibitory effects so that the production rate term of the primary biomarkers was positive. The value of E_max_ was set as 1 or 10 for stimulatory effects. EC_50_ was tested for 100 ng/mL or 1000 ng/mL to simulate different magnitudes of drug effects.

#### 2.3.4. Observation Duration

We performed SSEs for study durations of 3, 6, 12, and 24 h after administration of three ascending doses to compare the identification performance of MoA and the precision of drug-specific parameters with different observation durations. Both a large drug effect (EC_50_ = 100 ng/mL) and a small drug effect (EC_50_ = 100 µg/mL) were tested.

#### 2.3.5. Number of Animals

To determine how the number of animals used in the hemodynamic studies influences the identification of MoA and precision of parameter estimation, SSEs using three, four, or five animals were conducted. Both a large drug effect (EC_50_ = 100 ng/mL) and a small drug effect (EC_50_ = 100 µg/mL) were tested.

## 3. Results

### 3.1. Identifiability of Drug- and System-Specific Parameters

The mean relative prediction errors of drug- and system-specific parameters for both studied dosing regimens showed minimal bias (relative prediction error < 20%), suggesting that all the system- and drug-specific parameters can be adequately estimated (Figure 3). Depending on the action of the drug effect, K_out_ parameters, CO_0_, and HR_SV_ frequently showed an increased prediction error of up to 20%, in comparison to other parameters’ prediction errors of <5%. Inclusion of CO data, besides HR and MAP data, decreases the prediction errors, albeit not substantially. Not all test models could minimize successfully. The model refitted to the data including CO observations showed superior overall stability of estimations with a significantly (*p* < 0.05) higher percentage of successful minimizations compared to simulations that excluded CO observations. It was found that the median of the relative prediction error of parameters in SSE with five ascending doses was closer to 0 compared to the ones in SSE with three ascending doses (results on file); this indicates that the study design of five ascending doses could help to improve the accuracy of parameter estimation. Finally, we compared the final estimates of EC_50_ of models with unfixed and fixed system-specific parameters. The relative prediction errors of EC_50_ were found within ±10%, which indicates that fixing system-specific parameters has no significant influence on the estimation of EC_50_. (Appendix A); this finding may be considered when applying the model for analysis of hemodynamic studies.

### 3.2. Drug Effect Magnitudes

We found that for all simulated drug effect magnitudes and effect sites (i.e., HR, SV, TPR), the correct MoA could always be identified using a standard study design, i.e., for all re-estimated runs, substantially lower ΔOFV values were consistently observed for the correct model in comparison to alternative models with an incorrect MoA (results on file). These results confirm the utility of the CVS model to correctly identify drug MoA using standard study designs.

### 3.3. Observation Duration

A variable impact was observed of the duration of experimental observations on the correct identification of drug effect sites for high EC_50_ values (100 ug/mL), i.e., when the drug effect is close to the circadian variation (Figure 4). In contrast, for effects on HR an observation duration of 6 h is sufficient to identify the correct MoA; identification of the correct MoA for TPR and SV is only possible with an intermediate or low percentage of runs. Here, addition of CO data substantially increases the models’ ability to identify the correct MoA. When considering the relative prediction error for EC_50_ for these scenarios, we found that the extent of bias was correlated with the ability to identify the correct drug effect, with substantial bias in the EC_50_ for TPR and SV (50–100%) for shorter study durations; there was also a positive effect in reducing bias (Figure 5). The added value of including CO in terms of bias was less distinct. In contrast, when a lower EC_50_ value is used (Figure 6), the model can identify the correct mode-of-action for all drug MoAs; this is true even for a minimal study duration of 3 h, with only minimal bias in the EC_50_ of <5% (Appendix A). These findings highlight the ability of this model to infer hemodynamic MoA for short observation times and even to detect modest drug effects.

### 3.4. Number of Animals

A distinct effect of sample size (ranging between 3 to 5 animals) was found for different MoA for high EC_50_ values (100 ug/mL). For drug effect on HR, for all sample sizes the correct MoA could always be identified, whereas for SV and TPR the ability to identify a correct drug effect was much lower (Figure 7). Increasing both the number of animals and the additional measurement of CO substantially increased the model’s ability to identify the MoA for SV and TPR. Similar trends were observed for the relative prediction error distributions (Figure 8). When a lower EC_50_ value is used (Figure 6), the model can identify the correct mode-of-action for all drug MoAs for all sample sizes studied, with only minimal bias in the EC_50_ (Appendix A). These findings highlight the ability of this model to infer hemodynamic MoA for short observation times and even to detect modest drug effects.

## 4. Discussion

In this analysis, we demonstrated the utility of the CVS model framework for inference of drug- and system-specific parameters and identifying the correct MoA for different drug effect magnitudes and study design choices through systematic simulation and re-estimation-based practical identifiability analyses.

In our simulation analyses, we found that for commonly used experimental study designs, system- and drug-specific parameters could be identified with acceptable parameter estimation precision and bias, based on single-drug studies. We also found that the availability of additional dose levels (i.e., 3 versus 5 dose levels as tested) improved the stability and precision of the estimation procedure. Alternatively, as described previously by Snelder et al. [6,7], fitting datasets across multiple drugs may aid in more reliable estimation of system-specific parameters. Once such systems parameters are established, e.g., for a specific experimental setting, the systems parameters may be fixed to make the model more robust in identifying MoA and drug effect parameters, when applying the model to other drugs or drug candidates

We demonstrated how experimental study designs, such as duration of experimental observations and the sample size, can substantially influence the model’s ability to correctly identify drug MoA and estimate the associated drug effect parameters. In particular, when drug effects have a more modest magnitude, there is a substantial impact from the duration of observations and sample size. Importantly, for more modest drug effects, we showed that a model-based approach can still characterize the hemodynamic drug effects effectively. From our analyses, it is also clear that the power to identify drug effects at specific effect sites, i.e., HR, SV, and TPR, also varies substantially. Because the circadian rhythm magnitudes and residual errors of these variables are similar, it is likely this difference from the substantially different k_out_ values for these parameters is related to the magnitude of the circadian rhythm effects for these parameters relative to the drug effects (Figure 6). In this context, the identification of drug effects on SV was found to be most challenging, while inclusion of CO measurements dramatically improves identification of SV-associated drug effect. In addition, extension of the observation duration (Figure 5) or increasing the amount of animals was shown to enhance ability to identify SV-associated drug effects.

Our results show how design considerations, such as dose, sample size, and observation duration, all impact the estimation performance of hemodynamic drug effects and other model parameters. To this end, model-informed optimization of experimental designs is a relevant approach to ensure that both optimally informative experiments are performed and the amount of animals required can be minimized for ethical considerations [10,11,12]. Since intensive sampling of hemodynamic variables with telemetry is common in pre-clinical experiments, especially for measurements of HR and MAP, comparison of model performance with intensive and sparse sampling designs was not considered in our analysis. However, achieving a dense sampling frequency of hemodynamic variables can be more challenging to achieve, in particular in the clinic. To further explore the impact of sampling frequency in clinical contexts on parameter estimation and identification of mode-of-action, potential study designs can be evaluated using stochastic simulation–estimation strategies using the CVS model in the future, similar to the workflow described in the current report. Alternatively, the use of optimal design methodologies, such as D-optimal design, could be a relevant and more efficient strategy for evaluation and optimization of pre-clinical and clinical hemodynamic studies.

We found that CO data may not be required for the identification of MoA. This is important because CO is often measured invasively using arterial cannulation and a pulmonary artery catheter placement, which makes CO more complex to measure in comparison to HR and MAP data. The CVS model can, thus, be practically convenient for studying MoA in drug development. However, if available, CO data may be of value since we also found that the inclusion of CO data improved the precision of the parameter estimates and increased the rate of successful minimization in the second SSE analyses. In recent years, impedance cardiography has been used as an accurate non-invasive and lower-cost approach for obtaining frequent serial measurements of CO in clinical contexts. This may potentially benefit the use of the models in hemodynamic studies when the new technology becomes more routinely available in the future [13,14].

Although our analysis is fully based on simulation studies, we expect that our conclusions will also be applicable to experimental hemodynamic studies. Importantly, the CVS model used in our analyses has been developed based on extensive and densely sampled hemodynamic experiments for multiple drugs as previously described [7]; these experiments were able to accurately describe the observed hemodynamic responses as well as the circadian rhythms of cardiovascular variables, thereby forming the basis for the simulations in the current analysis. One factor not included in the current analysis is the potential situation when compounds have a dual mode-of-action, i.e., which affect multiple hemodynamic endpoints at the same time. In addition, we focused on drugs with a typical half-life of 4 h. For drugs with for example much longer half-lives, we expect that the estimation of k_out_ might be affected and could require extended study durations to achieve the same power to estimate the k_out_ and/or drug effect parameters. The parameter values used in this analysis were based on rat hemodynamic studies. However, we expect that our main conclusions can also be applied to other species if a species-specific model has previously been derived. Indeed, the hemodynamic model has recently been applied to dog hemodynamic studies [8].

In conclusion, we demonstrated the utility of the CVS model for identifying and quantifying drug MoA in pre-clinical cardiovascular studies. The MoA can be identified using the CVS model based on MAP and HR data, while the inclusion of CO data could further improve the model estimation. The findings in this study can also contribute to optimization of experimental study design and reduce the number of animals used in pre-clinical studies.

## Figures and Tables

**Figure 1 pharmaceutics-15-01175-f001:**
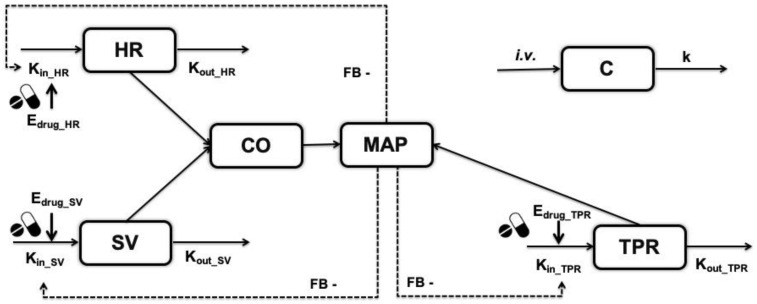
Model structure of the CVS model consisting of five hemodynamic variables of HR, SV, TPR, CO, and MAP and a one-compartment pharmacokinetic model of the hypothetical drug [7]. CO is defined as the product of HR and SV, while MAP is the product of CO and TPR. The drug affects the production rate of HR, SV, or TPR. C: drug concentration (ng/mL); HR: heart rate (beats/min); SV: stroke volume (mL/beat); TPR: total peripheral resistance (mmHg∙min/mL); CO: cardiac output (mL/min); MAP: mean arterial pressure (mmHg); FB: feedback effect of MAP (mmHg^−1^); E_drug_: drug effect on three primary effect sites. K_in_: production rate constant (K_in_HR_: beats*h^−1^/min, K_in_SV_: mL*h^−1^/beat, K_in_TPR_: mmHg*min*h^−1^/mL); K_out_: dissipation rate constant (h^−1^); k: elimination rate constant of the hypothetical drug (h^−1^).

**Figure 2 pharmaceutics-15-01175-f002:**
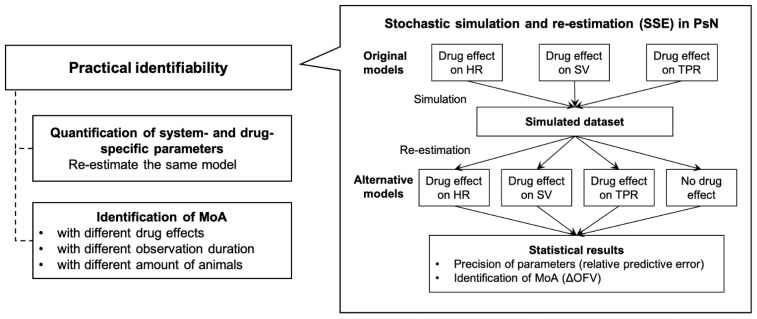
Overview identifiability analyses. SSE: stochastic simulation and re-estimation; PsN: Perl-speaks-NONMEM toolkit; HR: heart rate; SV: stroke volume; TPR: total peripheral resistance; MoA: mode-of-action; OFV: objective function value.

**Figure 3 pharmaceutics-15-01175-f003:**
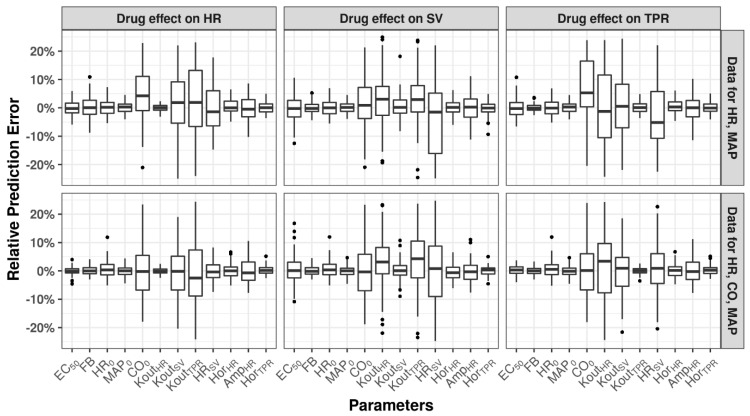
Identifiability of drug- and system-specific parameters. HR: heart rate; CO: cardiac output; SV: stroke volume; TPR: total peripheral resistance; MAP: mean arterial pressure. Relative prediction errors after simulation and re-estimation (SSE) analysis using observations for HR and MAP (**top**) or observations of HR, CO, and MAP (**bottom**) following three ascending doses. EC_50_: the concentration at which half of maximum effect is achieved; FB: regulatory feedback effect from MAP to HR, SV, and TPR; HR_0_: baseline value of HR; MAP_0_: baseline value of MAP; CO_0_: baseline value of CO; Kout_HR_: dissipation rate of HR; Kout_SV_: dissipation rate of SV; Kout_TPR_: dissipation rate of TPR; HR_SV_: direct effect constant between HR and SV; Hor_HR_: horizontal displacement of circadian rhythm of HR; Amp_HR_: amplitude of circadian rhythm of HR; Hor_TPR_: horizontal displacement of circadian rhythm of TPR.

**Figure 4 pharmaceutics-15-01175-f004:**
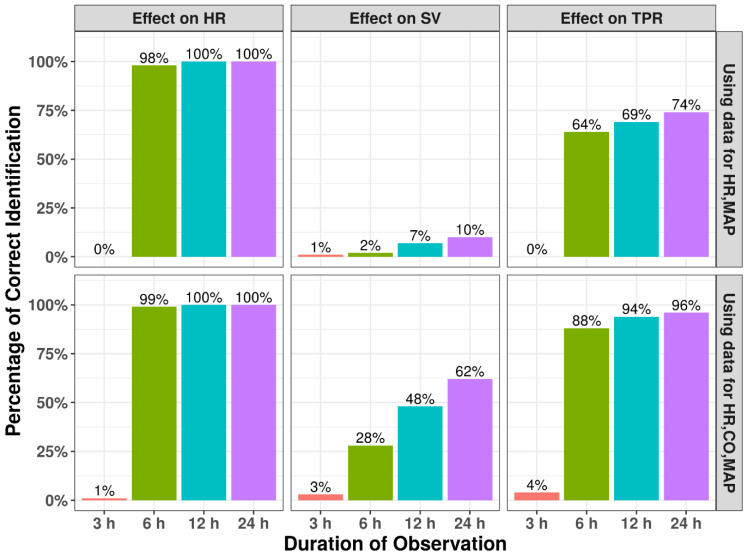
Impact of duration of experimental observations on the percentage of runs with correct identification of mechanisms of action. Results from simulation and re-estimation analyses using either observations of HR and MAP or observations of HR, CO, and MAP within 3 h, 6 h, 12 h, and 24 h. Emax was fixed as −1 and EC_50_ was fixed as 100 ug/mL in the original model. Emax was fixed to −1 and EC_50_ was fixed to 100 ug/mL in the original model. Correct identification is based on applying a likelihood ratio test compared to the base (no drug) model.

**Figure 5 pharmaceutics-15-01175-f005:**
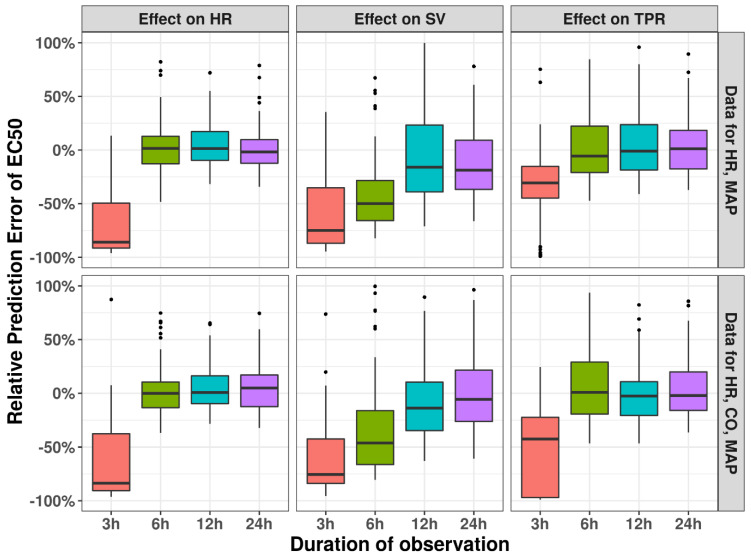
Impact of duration of experimental observations on relative prediction error of drug effect estimate EC_50_. Results from simulation and re-estimation analyses using either observations of HR and MAP or observations of HR, CO, and MAP within 3 h, 6 h, 12 h, and 24 h, while Emax was fixed as −1 and EC_50_ was fixed as 100 ug/mL in the original model.

**Figure 6 pharmaceutics-15-01175-f006:**
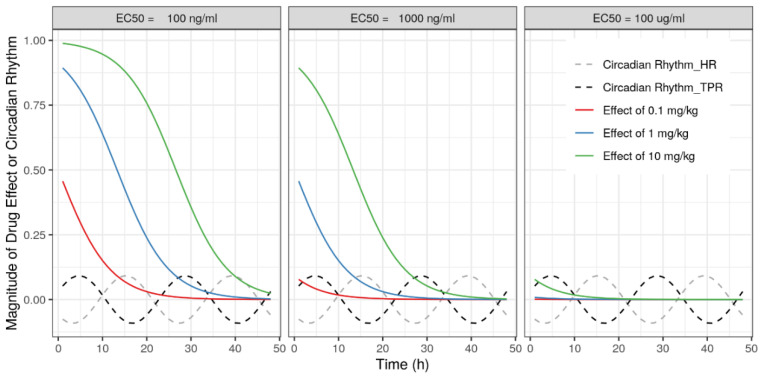
Comparison of magnitudes of circadian rhythm on HR or TPR and magnitudes of drug effect. Comparison of EC_50_ as 100 ng/mL (**left**), 1000 ng/mL (**middle**), and 100 ug/mL (**right**). Dark blue solid line: drug effect following a dose of 0.1 mg/kg; orange solid line: drug effect following a dose of 1 mg/kg; green solid line: drug effect following a dose of 10 mg/kg; black dashed line: magnitude of circadian rhythm effect of HR; gray dashed line: circadian rhythm effect of TPR.

**Figure 7 pharmaceutics-15-01175-f007:**
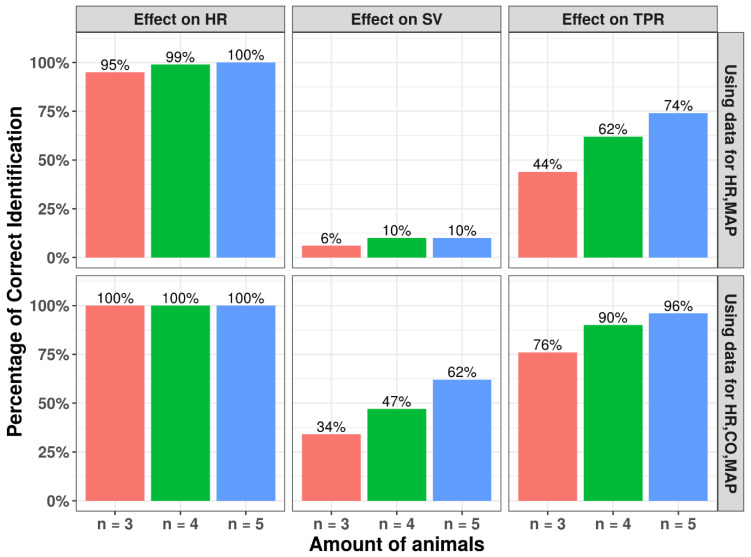
Impact of number of animals used in experiments on percentage of runs with correct identification of mechanism of action. Derived using SSE analyses using data for 3, 4, or 5 animals with observations of HR and MAP or observations of HR, CO, and MAP, while Emax was fixed as −1 and EC_50_ was fixed as 100 ug/mL in original model. Correct identification is based on applying a likelihood ratio test compared to the base (no drug) model.

**Figure 8 pharmaceutics-15-01175-f008:**
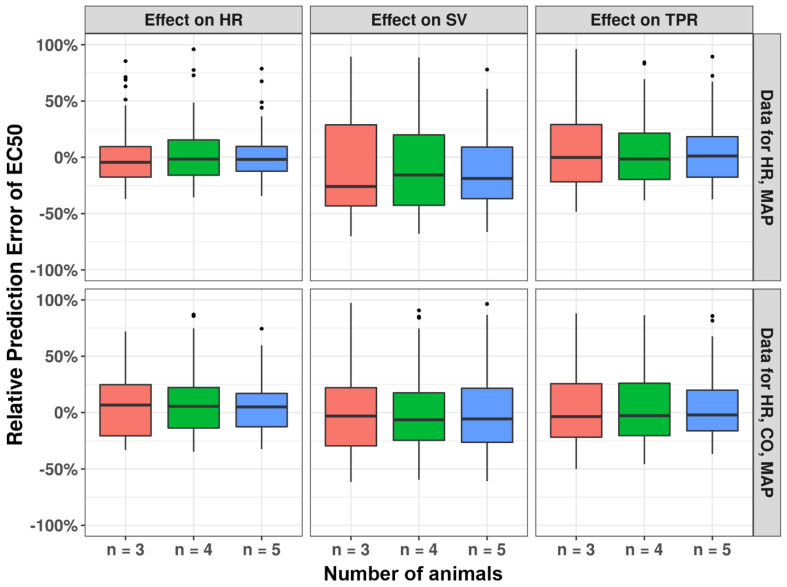
Impact of number of animals used in experiments on relative prediction error of drug effect estimate EC_50_. Derived using SSE analyses using data for 3, 4, or 5 animals with observations of HR and MAP or observations of HR, CO, and MAP, while Emax was fixed as −1 and EC_50_ was fixed as 100 ug/mL in the original model.

## Data Availability

Raw data is available upon request.

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
