# Peer review of "Evaluation of a Cardiovascular Systems Model for Design and Analysis of Hemodynamic Safety Studies"

_pharmaceutics, 2023, doi:10.3390/pharmaceutics15041175_

Round 1
Reviewer 1 Report
Here the authors present an analysis of a previously developed hemodynamic systems model. The authors carry out a simulation study using the model to examine its identifiability, model estimation performance, and impact of different study design and drug properties.
The work is interesting, and the simulation findings provide some meaningful results. I am perhaps missing reasoning around the model behavior and why these trends are seen. Also, it would be of value to take a more systematic approach to this analysis. Comments follow below.
General comments:
C1:
It would be interesting if the authors’ could share their insights into how these model exercises are used in practice. Translation is mentioned but not elaborated upon. For example, what are we aiming for here? Classification or quantification/both? This will impact our inference of the results you present here.
C2:
In earlier work the authors carried out model development, optimisation and validation against observed data. The development process is not quite clear here. Why carry out extensive development and validation to then carry out identifiability analysis and model verification, or am I misunderstanding the context? This is not clear.
C3:
Since the structural model and its parameter values have been established, it would be of interest if the authors could elaborate on the reasons for their findings in this study in the Discussion. For instance, is the challenge of estimating the effect on SV expected, why does this relate to the production rate of SV or the interaction with CO?
C4:
Related to the above comment - It would be informative to carry out a sensitivity analysis on this model (also according to different outputs), has this been done previously? If so, did the exercise agree with these simulations?
C5:
Going forward, how useful would D-optimal design be in this case? There are quite a few feedback loops indicated in the model diagram.
Or, is it sufficient to recommend a minimum sampling interval?
Minor comments:
C6: Page 4, line 128: “OFV… P<0.05”
Will this not make this approach susceptible to type 1 errors? Is this how model selection would be carried out in practice?
C7: Figure 1
It is worth clarifying this a bit. For example, the direction of the interaction between CO, HR and SV. C=drug concentration.
Author Response
Referee 1:
Here the authors present an analysis of a previously developed hemodynamic systems model. The authors carry out a simulation study using the model to examine its identifiability, model estimation performance, and impact of different study design and drug properties. The work is interesting, and the simulation findings provide some meaningful results. I am perhaps missing reasoning around the model behavior and why these trends are seen. Also, it would be of value to take a more systematic approach to this analysis. Comments follow below.
Author Response: Thank you for the helpful comments.
General comments:
C1: It would be interesting if the authors’ could share their insights into how these model exercises are used in practice. Translation is mentioned but not elaborated upon. For example, what are we aiming for here? Classification or quantification/both? This will impact our inference of the results you present here.
Author Response: We have expanded the introduction with the following paragraph: “As such, the CVS model has potential value to support classification of the drug MoA and the quantification of its effect size, to support preclinical study design optimization, and for its potential value to produce translational cross-species predictions, such as supporting dose selection in first-in-human studies.”
C2: In earlier work the authors carried out model development, optimisation and validation against observed data. The development process is not quite clear here. Why carry out extensive development and validation to then carry out identifiability analysis and model verification, or am I misunderstanding the context? This is not clear.
Author Response: The original model was developed based on a set of experimental data. However, when this model is applied for novel investigational compounds, it becomes important to characterize its performance, as the specific study designs used (available data) and the drug effect size present will determine its performance in terms of the correct and unbiased estimation of drug effects. To clarify, we have expanded the introduction with the following text: “the CVS model has potential value to support classification of the drug MoA and the quantification of its effect size, to support preclinical study design optimization, and for its potential value to produce translational cross-species predictions, such as supporting dose selection in first-in-human studies. The CVS model was developed based on an experimental data for a selected set of compounds with specific characteristics. More broad application of the CVS model is currently limited by of a lack of general quantitative characterization of the model performance in terms of correct identification and unbiased estimation of hemodynamic drug effects, given study design considerations such as the effect of differences in the available hemodynamic readouts, sample size, drug doses, and the magnitude of the effect size.”
C3: Since the structural model and its parameter values have been established, it would be of interest if the authors could elaborate on the reasons for their findings in this study in the Discussion. For instance, is the challenge of estimating the effect on SV expected, why does this relate to the production rate of SV or the interaction with CO?
Author Response: To clarify this point, we have expanded the discussion as follows: From our analyses, it is moreover clear that the power to identify drug effects at specific effect sites, i.e., HR, SV, TPR, also varies substantially. Because the circadian rhythm magnitudes and residual errors of these variables are similar, it is likely this difference from the substantially different kout values for these parameters in relationship to the magnitude of the circadian rhythm effects for these parameters relative to the drug effects (Figure 6). In this context, the identification of drug effects on SV was found to be most challenging, where inclusion of CO measurements dramatically improves identification of SV-associated drug effect. In addition, extension of the observation duration (Figure 5) or increasing the amount of animals was shown to enhance ability to identify SV-associated drug effects.
C4: Related to the above comment - It would be informative to carry out a sensitivity analysis on this model (also according to different outputs), has this been done previously? If so, did the exercise agree with these simulations?
Author Response: A sensitivity analysis could provide further insight into the relative importance of different parameters in this model, i.e., it relates to the intrinsic properties of the model. We believe such an analysis does not contribute to the goal of the current manuscript, which was to determine the performance of the CVS model to correctly (re-) estimate key parameter values, given different simulated study designs. To this end we believe it would be preferable not to expand our manuscript with the results of a sensitivity analysis.
C5: Going forward, how useful would D-optimal design be in this case? There are quite a few feedback loops indicated in the model diagram. Or, is it sufficient to recommend a minimum sampling interval?
Author Response: We fully agree that optimal design could be very relevant for the practical optimization of study designs for this model. We have added the following text to the discussion: “Our results show how design considerations such as dose, sample size, and ob-servation duration, all impact the estimation performance of hemodynamic drug effects and other model parameters. To this end, model-informed optimization of experimental designs is a relevant approach to ensure optimally informative experiments are per-formed, and to ensure the amount of animals requires can be minimized for ethical considerations10–12. Similar to the described workflow in the current report, potential study designs can either be evaluated using stochastic simulation-estimation strategies using the CVS model. Alternatively, the use of optimal design methodologies, such as D-optimal design, could be a relevant and more efficient strategy for evaluation and optimization of preclinical hemodynamic studies.”
Minor comments:
C6: Page 4, line 128: “OFV… P<0.05” Will this not make this approach susceptible to type 1 errors? Is this how model selection would be carried out in practice?
Author Response: Although we agree that in general a more conservative P-value could be appropriate. However, for the simulations in this manuscript we observed decreases in the OFV (-2LL) much larger magnitudes than those associated with P<0.05, indicating that for our analysis the risk for type 1 errors is negligible.
C7: Figure 1. It is worth clarifying this a bit. For example, the direction of the interaction between CO, HR and SV. C=drug concentration.
Author Response: We agree and have made clarifying changes in Figure 1.
Reviewer 2 Report
Fu et al present an interesting robustness and identifiability analysis of an earlier published cardiovascular toxicity predictive model from the same group. The results clearly indicate the usefulness of the model, and the inclusion of the model code encourages other groups to reproduce their results and apply the model in real-world pre-clinical experiments, which I can only encourage. The work is presented well and apart from some small typos (artrial instead of arterial at some places in the text, captions of Figures 6 and 7 change font size throughout the caption,... I recommend an extra readthrough to make sure the final version is up to snuff), I think it can be accepted in present form, given the small comments below.
1) Can the authors expand on why they only focussed on non-Qt related cardiovascular toxicity? It might be good to expand the introduction a bit to explain this (now it only states that is was not considered).
2) Can the authors expand the discussion a bit including why they assume that the performance of the model to predict SV is so much worse than the other outcomes? Are their experimental settings that improve this prediction (even longer experiments, more animals,...)
3) The authors switch between active and passive tense throughout the manuscript. I would recommend them to stay consistent with one.
Author Response
Fu et al present an interesting robustness and identifiability analysis of an earlier published cardiovascular toxicity predictive model from the same group. The results clearly indicate the usefulness of the model, and the inclusion of the model code encourages other groups to reproduce their results and apply the model in real-world pre-clinical experiments, which I can only encourage. The work is presented well and apart from some small typos (artrial instead of arterial at some places in the text, captions of Figures 6 and 7 change font size throughout the caption,... I recommend an extra readthrough to make sure the final version is up to snuff), I think it can be accepted in present form, given the small comments below.
Author Response: Thanks for your kind comments, we have corrected all the typos.
1) Can the authors expand on why they only focused on non-Qt related cardiovascular toxicity? It might be good to expand the introduction a bit to explain this (now it only states that is was not considered).
Author Response: We agree. We have added the following clarifying text in the introduction: “In this context, preclinical- and clinical cardiovascular safety studies and associated quantitative analyses have primarily focussed on early identification of QT prolongation effects, associated with an increased risk of Torsade de Pointes. However, drugs may also be associated with other non-QT cardiovascular hemodynamic drug effects, and such effects are now increasingly being considered in cardiovascular safety studies”
2) Can the authors expand the discussion a bit including why they assume that the performance of the model to predict SV is so much worse than the other outcomes? Are their experimental settings that improve this prediction (even longer experiments, more animals,...)
Author Response: This is not an assumption but simply a result from our simulations. We have expanded the discussion on this result including potential choices which can improve the ability to identify SV-related drug effects as follows: “From our analyses, it is moreover clear that the power to identify drug effects at specific effect sites, i.e., HR, SV, TPR, also varies substantially. Because the circadian rhythm magnitudes and residual errors of these variables are similar, it is likely this difference from the substantially different kout values for these parameters in relationship to the magnitude of the circadian rhythm effects for these parameters relative to the drug effects (Figure 6). In this context, the identification of drug effects on SV was found to be most challenging, where inclusion of CO measurements dramatically improves identification of SV-associated drug effect. In addition, extension of the observation duration (Figure 5) or increasing the amount of animals was shown to enhance ability to identify SV-associated drug effects”
3) The authors switch between active and passive tense throughout the manuscript. I would recommend them to stay consistent with one.
Author Response: Thanks for your comment. We have now made corrections to ensure consistency in the tenses used.
Round 2
Reviewer 2 Report
Thank you for updating the paper, I have no further comments.